# Needs, Barriers and Facilitators of Adolescents Participating in a Lifestyle Promotion Program in Oncology: Stakeholders, Adolescents and Parents’ Perspective

**DOI:** 10.3390/children9091340

**Published:** 2022-09-01

**Authors:** Johanne Kerba, Catherine Demers, Véronique Bélanger, Mélanie Napartuk, Isabelle Bouchard, Caroline Meloche, Sophia Morel, Nicolas Prud’homme, Isabelle Gélinas, Johanne Higgins, Daniel Curnier, Serge Sultan, Caroline Laverdière, Daniel Sinnett, Valérie Marcil

**Affiliations:** 1Research Center of the CHU Sainte-Justine, Montreal, QC H3T 1C5, Canada; 2Department of Nutrition, Université de Montreal, Montreal, QC H3T 1A8, Canada; 3Cardiometabolic Health, Diabetes, and Obesity Research Network (CMDO), Montreal, QC J1H 5N4, Canada; 4School of Physical and Occupational Therapy, McGill University, Montreal, QC H3G 1Y5, Canada; 5Institute of Nutrition and Functional Food, Quebec City, QC G1V 0A6, Canada; 6Division of Hematology-Oncology, CHU Sainte-Justine, Montreal, QC H3T 1C5, Canada; 7School of Rehabilitation, Université de Montréal, Montreal, QC H3N 1X7, Canada; 8School of Kinesiology and Physical Activity Sciences, Université de Montréal, Montreal, QC H3T 1J4, Canada; 9Department of Psychology, Université de Montréal, Montreal, QC H3C 3J7, Canada; 10Department of Pediatrics, Université de Montréal, Montreal, QC H3T 1C5, Canada

**Keywords:** adolescent, cancer, health promotion intervention, nutrition, physical activity, mental health, needs, barriers, facilitators, lifestyle

## Abstract

Treatments for adolescent cancer can cause debilitating side effects in the short- and long-term such as nausea and malnutrition but also cardiometabolic disturbances. Although the risk for cardiometabolic complications is greater for adolescents with cancer than younger ones, adolescents typically respond poorly to family-oriented health promotion programs. This study aims to assess the needs, barriers and facilitators to healthy lifestyle promotion interventions for adolescents with cancer and how to best adapt these interventions for them. Interviews were held with adolescents treated for cancer (*n* = 9) and parents (*n* = 6), focus groups were conducted with stakeholders working in oncology (*n* = 12) and self-report questionnaires were sent to stakeholders involved in a health promotion intervention (*n* = 6). At the time of interview, mean age of adolescent participants (40% female) was 17.0 ± 1.9 years (mean age at diagnosis: 14.6 ± 1.6 years). Verbatim and responses to questionnaires were coded and analyzed using qualitative methods. Stakeholder stated that adolescents with cancer need to access activities adapted to their age, to communicate with peers going through a similar experience, and to preserve their schooling and friendships. Barriers to intervention reported by adolescents, parents and stakeholders include lack of motivation, schedule conflicts, fatigue and treatment side effects. Some of the barriers mentioned by adolescents and parents include pain, post-surgery problems, school, physical deconditioning, and lack of time. Facilitators mentioned by adolescents and parents comprise trust in stakeholders’ expertise, personalized approaches, scheduling flexibility. Stakeholders recommended to build trust in the relationship, favoring non-moralizing teachings, adapt interventions to adolescents’ limited attention span and avoiding the use of long-term health benefits as a motivator.

## 1. Introduction

In North America, the overall 5-year survival rates for cancer in children (0 to 14 years), adolescents (15 to 19 years), and the adolescent and young adult (AYA) age group (15 to 39 years) range from 83 to 86% [1,2,3,4]. However, survival rates for several cancers vary considerably according to age [3,5]. One striking example is leukemia, with a 5-year survival of 85.6% in children (<15 years) compared to only 65.8% in AYAs [6]. The lower survival rate observed for some cancer types may be explained in part by the distinctive biologic characteristics of AYA cancers [5,7], often coupled with a need for more intensive treatment or more targeted therapy strategy [8,9,10,11,12,13]. In addition to survivorship issues, cancer treatments cause debilitating acute side effects including fatigue, pain, nausea, vomiting, diarrhea, and malnutrition [14,15,16]. Antineoplastic treatments can also lead to cardiometabolic disturbances including weight gain, insulin resistance, dyslipidemia, and high blood pressure [17,18,19,20,21]. Compared to children, adolescents’ cardiometabolic health is particularly affected by treatments, and these complications can persist after treatment in the short- and the long-term [17,19,20,22,23,24]. Accordingly, survivors of childhood and adolescent cancer are at higher risk of early mortality, mainly due to a second neoplasm or to a cardiovascular event, compared to the general population without a cancer history [23,25,26,27,28] or to siblings [29,30]. Although the exact mechanisms underlying the development of complications during and after treatment are unknown, several hypotheses have been proposed. Among them, lifestyle habits most probably play a key role [31,32,33,34]. Cancer treatments can impact appetite and cause dysgeusia, which may alter food habits and preferences [35,36,37]. This may result in cancer patients preferring carbohydrate-based, fatty, savory and salty foods, and in having lower intakes of fruits and vegetables [36,38,39]. Concurrently, during treatment period, patients engage in less physical activity (PA), leading to a more sedentary lifestyle and loss of muscle mass [40] and also present a lower physical fitness for the same amount of PA than healthy controls [41]. Cancer and treatment side effects can also impact the psychological health and quality of life of adolescents with cancer and survivors [42,43,44,45,46], further affecting healthy lifestyle habits [32,47].

It is widely recognized that adolescents have specific health and personal needs related to this period of life marked with multiple transitions [48,49,50]. More efforts are needed to address the unmet health care needs of adolescents as they may result in less favorable long-term health [51]. While several health promotion programs or prevention and educational interventions have been developed specifically for the adolescent population, those targeting precisely adolescents battling cancer are scarce. To our knowledge, the only one published consists of an 8-module Facebook-based educational intervention for adolescents treated for cancer [52]. Otherwise, all health promotion interventions for adolescents implemented during cancer treatments included children of all ages or young adults [53,54,55,56,57].

Changes in lifestyle habits in children and adolescents with cancer may occur early after the initiation of treatments [58,59,60]. Despite this knowledge, health promotion interventions in oncology were mainly held after the end of treatment period or during survivorship [61,62]. Health promotion interventions that begin early after cancer diagnosis may be a key element to establish healthy lifestyle habits and prevent cancer late effects.

Our team has tested the feasibility of implementing an early nutritional intervention in pediatric oncology, and we showed that adolescents had a lower participation rate than children [54]. The reasons explaining the lower engagement of adolescents are not clear, but we have suggested that it could partly be explained to the family-based design of the intervention which may not be adapted to adolescents’ preferences and needs [54]. Outside the field of oncology, many health prevention and promotion programs designed exclusively for adolescents were school-based as it was shown to be an effective strategy for the adolescent population [63]. In oncology, adolescents with cancer have reported different problems and concerns than younger children [64], but whether this is also true in the context of health promotion interventions is unknown. Cancer has a huge impact on the lives of adolescents, and they have specific needs that should be addressed. Therefore, before developing health promotion interventions for adolescents with cancer, it is important to understand the factors that will lead to better participation and engagement. This study aims to assess the needs, barriers and facilitators to healthy lifestyle promotion interventions for adolescents in oncology. The secondary objective is to investigate how to best adapt lifestyle promotion interventions for adolescents treated for cancer.

## 2. Methods

### 2.1. Study Framework and Ethics

This research is part of the VIE (Valorization, Implication, Education) study (web link: https://www.chusj.org/en/soins-services/H/Hematologie/Projet-Vie accessed on 31 August 2022), a non-randomized controlled feasibility study conducted at the CHU Sainte-Justine (CHUSJ) in Montreal (Quebec, Canada). The proposed integrative and personalized program informs newly diagnosed patients and their families about the benefits of adopting early healthy lifestyles, particularly a balanced diet, PA, and stress management to improve quality of life and prevent the incidence of late effects. The design of the interventions has been described in detail [54,57,65]. Briefly, participants 21 years old or younger at diagnosis, treated with chemotherapy or radiotherapy and able to give informed consent (by parents or legal guardians) were invited to participate. Patients were excluded if they were not being treated with chemotherapy and/or radiotherapy. Eligible participants were enrolled between the fourth and twelve weeks after cancer diagnosis. At recruitment, participants underwent an initial evaluation in nutrition, PA and psychological health. Follow-up visits were planned every month by alternating nutrition/PA consults. In parallel, the psychosocial support program was deployed in vulnerable families. Comprehensive nutrition, PA and psychological health evaluations were performed one year after the beginning of the intervention and after the end of cancer treatment. The control group was recruited from patients who had completed their treatment using the same criteria as for the protocol intervention group. No intervention was offered to the control group participants other than standard clinical follow-ups. The current work consists of a qualitative study conducted with a subgroup of the VIE study adolescent participants, in addition to their parents, and stakeholders working in oncology at CHUSJ (health care and research personnel). Participants were recruited from January 2018 until December 2021. Data were collected using focus groups, questionnaires and interviews. This study was approved by the Ethics Review Committee of the CHUSJ (#2021-3129 and #2021-3278).

### 2.2. Participants

#### 2.2.1. Stakeholders

Two categories of stakeholders were included in the study: (1) healthcare professionals and hospital-based teachers working with adolescents in oncology at CHUSJ who participated in the focus groups and (2) research health personnel who delivered the VIE study intervention, comprised of research kinesiologists and nutritionists who were eligible to answer the self-report questionnaire. Stakeholders were eligible for participation in focus groups if they were: (1) a healthcare professional or a hospital-based teacher working with children and adolescents in oncology at CHUSJ and (2) exposed to the VIE study (health care personnel). Stakeholders were eligible to answer the self-report questionnaire if they: (1) were a member of the VIE research team, (2) had delivered the nutrition (nutritionists) or PA (kinesiologists) interventions.

#### 2.2.2. Adolescents

Inclusion criteria were: (1) being between 12 and 21 years old at cancer diagnosis; (2) having been treated with chemotherapy and/or radiotherapy; (3) being able to give informed consent (parent or legal guardian); and (4) having participated in the VIE study for a minimum of 1.5 years.

#### 2.2.3. Parents

Inclusion criteria were: (1) being the parent or legal guardian of an adolescent aged between 12 and 21 years at cancer diagnosis who participated in the VIE study for a minimum of 1.5 years and (2) being able to give informed consent.

### 2.3. Recruitment and Data Collection

#### 2.3.1. Focus Groups with Stakeholders

Participants (oncologist, nurse, nutritionist, rehabilitation professional, psychologist and hospital-based teacher) were approached by email. A total of 3 focus groups were held. Discussions were recorded using 2 digital recorder devices. The leader and moderator of the focus groups were two of the authors, J.K. and C.D., respectively. Neither of them delivered interventions to the VIE study participants. Two focus groups lasted 90 min each, of which 30 min of discussions were specific to adolescents with cancer. One focus group lasted 90 min and focused exclusively on adolescents treated in oncology. A total of 150 min of focus group verbatim covering adolescent-related content was transcribed and revised by J.K. and C.D. After the coding and analysis of the third focus group verbatim, data saturation was reached as most themes were recurrent with those from the 2 previous focus groups [66].

#### 2.3.2. Self-Report Questionnaires to VIE Study Stakeholders

The 4-question questionnaire was developed specifically for the VIE study by the research team. The purpose of this questionnaire was to obtain feedback from VIE study stakeholders on their perception of the needs of adolescents treated for cancer and how to optimize their participation in healthy lifestyle interventions. It was sent electronically to stakeholders through a survey transmission platform (Google Forms). Responses to the self-report questionnaires were anonymous.

#### 2.3.3. Interviews with Adolescents and Parents

All adolescents and parents participating in the VIE study were invited to meet the interviewer for a face-to-face interview. Two types of semi-structured interviews were conducted: barriers and facilitators interviews and exit interviews. We established a priori that participants could not take part in both types of interviews. A total of 31 in person interviews (*n* = 31), conducted individually or as an adolescent-parent dyad, were audio recorded. For the current study, only interviews conducted with adolescents and parents of adolescents were considered (*n* = 12), including 8 individual interviews about barriers and facilitators and 4 exit interviews. At the time interviews were conducted, all adolescent participants had completed their cancer treatments. Interview duration ranged from 8 to 32 min for a total of 194 min of verbatim. To reduce bias, all interviews were conducted by the same person (C.D.) who is an occupational therapist and doctorate candidate.

To assess the barriers and facilitators to participation in the VIE study, of a total of 8 individual interviews, 5 were conducted with adolescents and 3 with parents. Interviews were held between November 2020 to January 2021. Of the 3 parents interviewed, 2 were parents of teenagers who had themselves participated in a one-on-one interview about barriers and facilitators.

Exit interviews were intended to obtain participants’ feedback on the VIE study and suggestions to improve the program or the interventions. These interviews took place during their final assessment at the end of the study and were held between March 2021 and March 2022. A total of 4 exit interviews were conducted with 3 adolescent-parent dyads and with 1 adolescent only.

### 2.4. Data Analysis

Focus groups verbatim and answers to questionnaires were coded and analyzed by J.K. using a thematical analysis with the NVivo qualitative analysis software (version 1.6.1) and using an inductive approach based on the conceptual framework entitled: Consolidated Framework for Implementation Research (CFIR) [67]. The 5 domains of the CFIR construct were used as a guide to assess barriers and facilitators to participation: intervention characteristics, inner setting, outer setting, individual characteristics and implementation process [67]. A little more than 20% of the focus group verbatim was coded and analyzed by a second investigator (C.D.) for inter-judge validation. Coding of the answers to self-report questionnaires were integrated in the same coding tree as the focus groups as questions related to adolescents were identical in both methods. Main themes were identified separately by J.K. and C.D. and were discussed until an agreement was reached. Results were then presented to V.M. to ensure agreement on the main themes.

Interview verbatim was transcribed by C.D. and coded by J.K and C.D. A thematic analysis using deductive and inductive approaches were used to analyze barriers and facilitators’ interviews based on the CFIR framework [67]. Because there is no theoretical background or existing knowledge supporting the selection of themes with regard to questions from the exit interviews, an inductive approach was used [66]. For data presentation, all quotations have been translated from French to English as all focus groups and interviews were held in French.

## 3. Results

### 3.1. Characteristics of Participants

#### 3.1.1. Focus Groups

Participants included occupational therapists, physiotherapists, a nutritionist, a psychologist, a nurse, oncologists and hospital-based teachers working with adolescents in oncology at CHUSJ (Table 1). The number of participants per focus group varied between 3 and 5, for a total of 12 attending professionals.

#### 3.1.2. Self-Report Questionnaires

Questionnaires were sent to 9 stakeholders who had actively worked with patients, either in the nutrition or the PA components of the VIE study. Completed questionnaires (*n* = 6) were received from 2 nutritionists and 4 kinesiologists (Table 1).

#### 3.1.3. Interviews with Adolescents and Parents

A total of 9 adolescents participated in the interviews out of a possible 14 adolescents (Table 1). The reason for not participating was lack of time. The age at the interview varied between 13.8 and 19.9 years. In total, 6 parents participated in the interviews. Socio-demographic characteristics were collected from 5 parents as 1 did not complete the case report form.

### 3.2. Main Themes Related to Adolescents in Oncology

Qualitative data analysis of the focus groups, self-report questionnaires and interviews resulted in the identification of four main themes: (1) Specific needs of adolescents treated for cancer; (2) Barriers and facilitators to participating in lifestyle promotion interventions; (3) Impact of cancer and treatments on adolescents; and (4) suggestions on how to adapt healthy lifestyle promotion interventions for adolescents in oncology.

### 3.3. Specific Needs of Adolescents Treated for Cancer

Stakeholders agreed on the fact that adolescents should be considered as a distinct group with regard to healthcare services as their motivations and needs are different from those of children and adults. One of the recurrent topics mentioned by stakeholders was adolescents’ need for autonomy and freedom, and the need to test limits. One oncologist stated that adolescents need to have a sense of control and to make their own decisions. Stakeholders raised the importance of including adolescents in decision-making, including objective settings, instead of imposing on them the priorities of the study and stakeholders.

Specific to adolescents in oncology, many stakeholders mentioned that adolescents need to have access to services and interventions adapted to their age, to connect with other adolescents experiencing a similar situation, to regain a positive body image and to maintain their physical capacities. Stakeholders, including healthcare professionals, research personnel and hospital-based teachers expressed that adolescents treated for cancer have a need for independence, for their personal space, for time without their parents, to express their feelings and anger, to discover new enjoyable activities adapted to their new reality and medical restrictions. A comprehensive description of the perception of stakeholders regarding the needs of adolescents in oncology expressed during focus or in self-report questionnaires is presented in Table 2, including stakeholders’ quotations.

### 3.4. Barriers and Facilitators to Participation in Lifestyle Promotion Interventions

Several perceived barriers and facilitators to adolescents’ participation in healthy lifestyle promotion interventions were raised by stakeholders, adolescents and parents (Table 3). They were classified into the five categories of the CFIR framework: intervention characteristics, inner setting, outer setting, individual characteristics and implementation [67]. The most recurrent barriers were related to treatment side effects including fatigue, pain and nausea, lack of time and motivation, schedule conflicts and depression to name but a few. The main facilitators included trust in healthcare providers’ expertise, personalized approaches, scheduling flexibility, and respect of the adolescents’ limits by the stakeholders. The COVID-19 pandemic was identified both as a barrier and a facilitator. It was found to be beneficial by many parents and adolescents as they did not have to travel to the hospital, but it was also perceived as a barrier by some adolescents who were not interested in participating in remote interventions.

### 3.5. Impact of Cancer on Adolescents

Although this was not a specific study objective nor was it a precise question asked during focus groups or included in the self-report questionnaire, several additional insightful discussion topics raised during focus groups and were cited in the self-report questionnaire. Most of them were related to the multi-level impact of having a cancer diagnosis during adolescence. Stakeholders mentioned the physical consequences of cancer that often lead to a poor body image, but they also referred to the loss of physical capacities which leads to the loss of self-esteem. Stakeholders also reported the psychological and social consequences of having cancer during adolescence. They indicated that adolescents face many changes at the same time and that their world appears to be falling apart. Stakeholders also mentioned that adolescents being treated for cancer often have strained relationships with their parents and perceive a lack of understanding from their friends. Arising results pertaining to consequences of cancer mentioned by stakeholders during focus groups and in self-report questionnaires are highlighted in Table 4.

### 3.6. Suggestions to Adapt Oncology Lifestyle Promotion Interventions for Adolescents

Input from stakeholders, adolescents and parents has generated a rich body of suggestions on how to adapt lifestyle promotion interventions in oncology to adolescents’ reality and favor success (Table 5). The importance for the stakeholder to build trust in his relationship with the adolescent is a central emerging theme. Having a room dedicated to adolescents in oncology has been a central talking point during focus groups and was expressed as a simple means of improving adolescents’ quality of life during their hospital stay. The need to implement non-moralizing and individualized interventions was also addressed.

## 4. Discussion

This study highlights that, in the context of healthy lifestyle promotion interventions, adolescents battling cancer need to access activities and services adapted to their age, to confide, to express their feelings and anger, to communicate with other adolescents going through cancer and to maintain a sense of normalcy. Barriers to intervention include nausea, fatigue, pain, physical deconditioning, lack of motivation and schedule conflicts. Facilitators comprise stakeholders’ flexibility and availability, as well as their capacity to listen and understand adolescents’ difficulties. Other facilitators involve interventions being carried out during waiting times between medical appointments and remotely. The COVID-19 pandemic was perceived as both a barrier and a facilitator as remote interventions were alleged to be time-saving for some but demotivating for others. The reported impacts of cancer on adolescents include loss of physical capacities, low self-esteem, body image issues, lack of understanding from friends and loss of autonomy. The main recommendations for adapting health promotion interventions to adolescents in oncology pertain to the importance of building trust in the relationship with the teenager. Other recommendations include favoring non-moralizing teachings, building common objectives with the adolescent, and addressing their griefs before initiating activities.

Similar to our study, the need to connect with other adolescents living a comparable situation in addition to preserving friendships has been reported in children and AYAs during and after cancer treatments [68,69,70,71,72]. This consists of an important unmet need as adolescents and AYAs usually receive care in a pediatric hospital with younger children or in adult care settings [73,74,75,76]. This has been mentioned in a study exploring the needs of AYAs treated for cancer in adult surgery clinics in which they expressed their preference to be treated alongside patients their own age, rather than older adults [74]. In another study exploring the experience of survivors of AYA cancer (16 to 40 years old at diagnosis), it was stated that AYAs struggled to find their place in the health care system [77]. In order to fill these gaps and unmet needs, more multidisciplinary oncology care facilities are being developed specifically for the AYA population [78,79,80,81]. In a review article, key elements to consider when developing a program dedicated to AYAs with cancer included adopting a multidisciplinary approach with health care professionals specialized in AYA cancer and dedicating a space exclusively for AYAs to facilitate interactions and prevent isolation during hospitalization [76].

Maintaining a sense of normalcy is an important need identified in our study, but also in others [44,68,71,73]. We found that adolescents do not want to be perceived by others as being only cancer patients. This was described in two studies that performed semi-structured interviews: adolescents with cancer have negative reactions to sympathy and special treatment, they want to be perceived as normal teenagers and be treated as normal as possible [44,68]. A study exploring the preferences of AYAs early after cancer diagnosis revealed that spending time with healthy peers helped them feel more normal during the treatment period [71]. When examining the concept of friendship, it was mentioned that peers often lack the knowledge to treat adolescents with cancer as normal teenagers [73]. In our study, stakeholders mentioned the need for normalcy in multiple contexts: continuing schooling during hospitalization, preserving friendships and maintaining a routine for the adolescent and the rest of the family. Another emergent theme pertained to the importance of allowing adolescents enough time to decide whether they want to participate or not in a healthy lifestyle intervention. To our knowledge, it is the first time this concept has been reported in the context of adolescent oncology. In view of this finding and in consideration for the deployment of future health promotion interventions specific for adolescents, we suggest not to overly limit the recruitment period to allow them time to decide whether or not they want to participate. Offering the possibility to enroll in the intervention at different times during the treatment trajectory could also meet this need.

A study exploring the barriers to the wellness of survivors of pediatric cancer aged between 11 and 30 years and their families found, similar to us, that barriers to PA include fatigue, lack of strength, doctors limiting PA, poor motivation to exercise and depression [32]. Parents feeling torn between supporting their child undergoing cancer treatments and responding to the needs of their other children has also been reported in the literature [82]. In our study, parents did mention lack of time as a barrier to participation, but they also pointed out that the nutrition and PA interventions for their adolescent took weight off their shoulders and allowed them to have more time to take care of their other children.

Autonomy is an important value for adolescents [83]. In our study, stakeholders pinpointed adolescents’ desire for autonomy as a barrier to participation in health promotion interventions. This was also mentioned as a barrier to behavioral health interventions targeting healthy adolescents where stakeholders advise teenagers on their personal choices [83]. Conversely, adolescents’ need for autonomy has been channeled positively in a double-blind, randomized, controlled experiment involving more than 500 adolescents to motivate adolescents in adopting healthier food habits [83]. To our knowledge, some of the barriers to participation in PA sessions that have emerged from our study have not been outlined elsewhere in the literature. These include the timing of the beginning of the program, lack of motivation due to the COVID-19 pandemic and pride associated with the loss of physical capacities.

Most studies evaluating the needs of adolescents treated for cancer also included young adults up to 39 years of age [74,84,85,86] or younger children [64,87]. A qualitative study assessing the problems, stressors and needs of children and adolescents with cancer enrolled 520 participants between 10 and 18 years of age, of which 359 were aged between 13 and 18 years [64]. Participants reported multiple needs including those for emotional support from society, support from nurses and doctors and to access education while hospitalized [64].

Stakeholders participating in our focus groups underscored the impact of cancer on adolescents and the numerous troubles and griefs adolescents battling cancer may experience. In the literature, interviews with adolescents in oncology revealed that cancer was related to pain and treatment side effects, fear of cancer relapse, hospitalization, loneliness, lack of relationships with peers, stigmatization, stress, anxiety, and body image issues [73,88,89].

Our study has highlighted the necessity for building a strong bond to facilitate the communication and collaboration between the teenager and the stakeholders. The importance for healthcare professionals to build trust in the relationship with their patients was outlined in studies evaluating AYAs’ priorities and needs during cancer treatments, specifically having close contact and the opportunity of sharing their thoughts with their nurse [74,90].

As mentioned in our study, adolescents with cancer need to access activities and services adapted to their age and need to obtain information and clear answers to their questions, even after the treatment period has ended. Accordingly, in young adult survivors of childhood cancer, the main barriers to health care follow-up were the lack of knowledge related to late effects of primary care physicians [91,92]. Given that it is well documented that survivors of AYA cancer are at higher risk for cardiometabolic complications [26], there is a pressing need for health promotion interventions tailored for this age group to optimize their motivation, engagement and participation. Continuing medical education as well as community teachings should be put in place to ensure that healthcare professionals and adolescent survivors are aware of their health risks.

Based on the study results, we recommend that future health promotion interventions for adolescents with cancer consider assuring the stability of the stakeholders during the intervention in order to promote the establishment of a relationship of trust. Remote interventions should be made available in addition to face-to-face interventions. The option of continuing the intervention after treatment has ended should also be offered. Moreover, initiatives should be put in place to encourage interactions with adolescents experiencing similar situations and with healthy peers in order to break isolation and to maintain a sense of normalcy. In addition, during the development process of a health promotion intervention, we recommend including adolescent patients of different ages, with different diagnoses and at different stages in the treatment trajectory. Finally, because adolescents tend to live in the present moment, the short-term benefits of a healthy lifestyle should be emphasized rather than the long-term health benefits.

## 5. Study Strengths and Limitations

A strength of our study is having consulted with stakeholders working in oncology, adolescents treated for cancer and parents which allowed us to gather a comprehensive perspective. Combining three methods of data collection (interviews, focus groups, self-report questionnaires) contributes to the robustness of the results. By focusing only on adolescents rather than children of all ages or AYAs, our study filled a gap in knowledge in the field of adolescent oncology. Another strength is that half the parents were fathers, a population who is often underrepresented in qualitative studies involving parents of children with cancer [60,93]. Our study has some limitations. It is possible that conducting the study in the hospital where adolescent participants received their treatments and where stakeholders work resulted in a desirability bias. There is a possibility of selection bias since adolescents participating in the VIE study may be more receptive to healthy lifestyle interventions. Indeed, in our study, adolescents who refused to participate in the VIE study or who dropped out of the study were not included. Another limitation is the small sample size for all groups of participants. Lastly, given the monocentric study design, our results may not be generalizable to the general adolescent oncology population.

## 6. Conclusions

Developing tailored health promotion interventions for adolescents treated for cancer may optimize their motivation and interest in participating in such programs. Several facilitators to adolescents’ participation in health promotion interventions identified in our study were interconnected with the adolescents’ needs identified. Health promotion interventions represent a unique opportunity to increase adolescents’ knowledge on potential side effects and long-term risks of cancer treatments. Mostly, health promotion interventions could contribute to embedding early healthy lifestyle habits in adolescents treated for cancer. Adolescents’ particular needs must be recognized and taken into consideration to provide optimal care and maintain their well-being during the challenging period of cancer treatments and during survivorship.

## Figures and Tables

**Table 1 children-09-01340-t001:** Demographic characteristics of participants.

Stakeholders (*n* = 12)—Focus Groups	
Sex—female *n* (%)	8 (66.7)
Years of experience in respective profession *mean* (SD)	18.4 (10.4)
Years of experience in oncology *mean* (SD)	12.7 (8.1)
Discipline *n* (%)	
Nurse	1 (8.3)
Oncologist	2 (16.7)
Rehabilitation professional	5 (41.7)
Nutritionist	1 (8.3)
Psychologist	1 (8.3)
Hospital-based teacher	2 (16.7)
Stakeholders (*n* = 6)—Self-report questionnaires	
Sex—female *n* (%)	5 (83.3)
Years of experience in respective profession	7.8 (5.5)
Years of experience in oncology *mean* (SD)	4.3 (2.5)
Discipline *n* (%)	
Nutritionist	2 (33.3)
Kinesiologist	4 (66.7)
Adolescents (*n* = 9)—Interviews	
Sex—female *n* (%)	4 (40.0)
Age at moment of interview *mean* (SD)	17.0 (1.9)
Age at diagnosis *mean* (SD)	14.6 (1.6)
Diagnosis *n* (%)	
ALL	5 (55.6)
Lymphoma	2 (22.2)
Sarcoma	2 (22.2)
Parents (*n* = 6)—Interviews	
Sex—female *n* (%)	3 (50)
Ethnicity *n* (%)	*n* = 5
Caucasian/white	5 (100)
Marital status, *n* (%)	*n* = 5
Married/common-law partner	4 (80.0)
Separated/divorced/widower	1 (20.0)
Education level, *n* (%)	*n* = 5
Less than high school	0 (0)
High school	1 (20.0)
College	2 (40.0)
University	2 (40.0)
Approximative personal income, *n* (%)	*n* = 5
<$29,999	0 (0)
$30,000–$69,999	1 (20.0)
$70,000–$109,999	3 (60.0)
>$110,000	0 (0)
Prefers not to answer	1 (20.0)

Rehabilitation specialists include physiotherapists (*n* = 2) and occupational therapists (*n* = 3). ALL: acute lymphocytic leukemia. Lymphoma diagnosis includes lymphoma (*n* = 1) and Hodgkin lymphoma IVB (*n* = 1). Sarcoma diagnosis includes Ewing’s sarcoma (*n* = 1) and rhabdomyosarcoma (*n* = 1).

**Table 2 children-09-01340-t002:** Stakeholders’ perception of the specific needs of adolescents in oncology.

Needs	Stakeholders’ Quotations from Focus Groups and Self-Report Questionnaires
Need to access activities and services adapted to their age	*“You have to make a distinction between them [adolescents] and children. All of the activities are for young children. The clowns are extraordinary and there were plenty of other visits during the pandemic, but all of them were for children.”* (Hospital-based teacher, focus group) *“I remember my first year here, there was Cachou [a kangaroo mascot] who came to offer teddy bears to kids in oncology. Well, the mascot forced my big teenager to have her picture taken when she had just lost her hair... She was in secondary 5 [grade 11]…Another teacher who was with us at the time got so upset. You know, their body image is important, so when there are many visits and they take pictures…”* (Hospital-based teacher, focus group)
Need for autonomy and freedom	*“[They a need to] maintain as much as possible their personal autonomy.”* (Kinesiologist, self-report questionnaire) *“It**’**s weird because you start to have a certain degree of freedom and all of a sudden, people come and tell you: you have to eat this, you have to do this, you have to do that. It**’**s opposite to where they [the adolescents**]* *are at in their mind.”* (Hospital-based teacher, focus group)
Need to have a sense of control on their lives and need to make their own decisions	*“I think there are 2 keys to success with teenagers. Consistency is important: even if you [the stakeholders**]**don**’**t succeed the first time, at least they know that it is someone who cares about their well-being and who pays attention to them. So, it**’**s about giving them the opportunity [to make their own decisions] because they can**’**t decide anymore on most aspects [of their lives]. They have lost authority in their decision-making power and in all of their daily cares. There is something optional about participating [in health promotion interventions]. To have that decision is important to them.”* (Oncologist, focus group)
Need to continue activities that are important to them	*“One of our students was passionate about ice skating […]. All of the kinesiologists worked in this direction. She started skating again today, it**’**s amazing. But this is what motivated her towards her goal because, she was doing competitive ice skating.”* (Hospital-based teacher, focus group)
Need to maintain a sense of normalcy	*“It**’**s good to play video games all the time. But at one point, it just becomes boring, and you lose pleasure because it just isn**’**t normal…”* (Oncologist, focus group)
Need to keep a certain routine/ Need to maintain limits	*“I tell my patients, after the announcement of the diagnosis, whether they are younger or older: “well, look, we have the diagnosis, we will start the treatments, there are parts that will be easier than others.” But I tell parents: he/she remains your child, you know, you still have to continue to have a routine, well, a routine in quotation marks…and [you need to continue to keep] some discipline and a sense of normality as much as possible, not only for your child, but also for the rest of your family.”* (Oncologist, focus group) *Sometimes we tell them [adolescent patients]: “it doesn**’**t bother me if you sleep in the morning when I arrive [in your hospital room]. But if you are sleeping because you were playing video games until midnight, it**’**s not the same as if you sleep because you didn**’**t feel well the day before and if you felt sick. The idea is to put back a routine and all that.”* (Hospital-based teacher, focus group)
Need to communicate with other teens in the same situation	*“I think that yes, the young people would like to meet. You just have to facilitate that. It has to be easy. Because, indeed, there are some teens who say: “well I saw another one [oncology patient] of my age, but that**’**s all, we didn**’**t talk to each other…”[…] They**[**adolescents in oncology**]**miss being able to communicate, they miss it a lot.”* (Psychologist, focus group).
Need to preserve their schooling and friendships	*“It may sound silly, but let**’**s say a teenager doesn**’**t have to repeat a full school year and that they can come back in the same group of friends. Well, that is extremely important to them.”* (Oncologist, focus group)
Need to regain a positive body image/ Need to maintain physical capacities	*“[Adolescents treated in oncology need to] Maintain, define, and regain the integrity of their personal image associated with the changes of the body, but also with the decrease in [physical] capacities.”* (Kinesiologist, self-report questionnaire) *“With regards to treatment period, adolescents need to maintain their assets: strength, VO_2_ max, flexibility, motor skills, etc.”* (Kinesiologist, self-report questionnaire)
Need to confide	*“They [adolescents in oncology] miss being able to communicate, they miss it a lot.”* (Psychologist, focus group) *“They need to be able to be with and to talk to someone else. They may have their families, but when you are a teenager, you do not want to have to talk about that with your parents. Your parents are already aware of many aspects of your private life.”* (Hospital-based teacher, focus group)
Need time to make decision	*“I think I can count on the fingers of one hand, those [adolescent patients] who started [French or math classes at the hospital**]**right away and who were motivated, but we must persevere. We have one who had refused every day for 4 months. But now, when she comes [to the hospital] once a month, she still calls us**[**the 2 school-based teachers**]**. But that**’**s it, it can be long, you have to give them time.”* (Hospital-based teacher, focus group)
Need for independence/Need for space and time without their parents	*“Sometimes, there were parents who attended classes and they would answer for their child. I had one [adolescent] this week, who told his mother: “It**’**s not you who is at school, it**’**s me.”* (Hospital-based teacher, focus group)
Need time with stakeholder without parental presence	*“I often say: if it doesn**’**t bother you, I would like to ask him/her [alone**]**routine questions that I ask all teenagers. […] There are sometimes parents who are like: “no, no, it**’**s ok, we are aware of everything.” You believe them but sometimes, there is a certain discomfort.”* (Oncologist, focus group)
Need to express their feelings and their anger	*“[I tell them] It really bothers you, you are not in a good mood, no problem. […] I know today is a “shitty day.” Do you want to do it? [schooling] or you can**’**t do it? There**’**s no stress if you just don**’**t want to. Go ahead, I**’**m not your parent. Everything you tell me, doesn**’**t affect me. I still can sleep at night. I**’**m able to take it...”* (Hospital-based teacher, focus group)
Need for information	*“[Adolescents need to] Obtain information and practical knowledge of their new reality and receive clear answers to the numerous questions that are generating**[**in their mind**].**”* (Kinesiologist, self-report questionnaire)
Need to find enjoyable activities adapted to their new reality	*“[Adolescents tell me:] “I can no longer go outside, do this or that…” Many teenagers discovered that they like cooking during the nutrition workshops. […] I tell them: “There may be activities that you have not discovered yet, maybe I can give you some ideas of activities that other adolescents enjoy doing.”* (Psychologist, focus group) *[They need] to discover new activities, cool activities for teenagers, that are adapted to their actual condition. They may have never done it before. […] A yoga discovery workshop, a dance or a “zumba” discovery workshop. But the workshops must be adapted. You adapt an activity that the teenager can do or can start doing again. And you tell them how to do it* *[**to avoid injuries**]**.”* (Occupational therapist, focus group)

**Table 3 children-09-01340-t003:** Perceived barriers and facilitators to adolescents’ participation health promotion interventions in oncology.

Barriers				Facilitators			
Intervention characteristics	S	A	P	Intervention characteristics	S	A	P
Face-to-face interventions	X			Commitment and consistency of stakeholders	X		
One-on-one interventions	X			Stakeholders’ expertise and flexibility		X	X
Use of long-term health as a motivator	X			Personalized interventions	X	X	X
Focus on young children in pediatric hospital environment	X			Remote interventions due to the pandemic		X	X
Too much time between interventions			X	Interventions being held during waiting time between medical appointments		X	
Timing of the beginning of the project	X		X	Individualized approach and recommendations		X	X
Large number of interventions			X	Personalized advice from nutritionist and kinesiologist		X	X
Inner setting	S	A	P	Inner setting	S	A	P
Schedule conflicts	X	X	X	Encouragement from stakeholders	X	X	X
Poor timing of intervention appointments (versus chemotherapy)	X	X	X	Stakeholders being understanding towards adolescents’ difficulties	X	X	
				Availability and listening by stakeholders	X	X	
				Stakeholders perceived as nice persons to talk to		X	
				Respect from stakeholders when adolescent does not wish to participate	X	X	X
Outer setting	S	A	P	Outer setting	S	A	P
Distance between home and the hospital			X	Interventions perceived as taking weight off parents’ shoulders and allowing them to spend more time with their other children			X
School		X	X	Online schooling due to pandemic			X
Demotivation due to the pandemic/no interest for remote interventions	X	X	X	Parents being together as a couple		X	X
Preference for other type of activities (outside the hospital)		X	X	New knowledge acquired in nutrition and kinesiology		X	
Adolescents not feeling concerned with proposed interventions		X	X	Family support during treatment period		X	X
Parents being separated			X				
Lack of activities during the pandemic	X	X					
Pandemic preventing adolescents to be active and reaching out to others		X	X				
Characteristics of individuals	S	A	P	Characteristics of individuals	S	A	P
Medical contraindication to physical activity	X	X	X	Improvement in adolescents’ self-confidence	X		
Desire for autonomy	X		X	Adolescents finding interventions fun/pleasant		X	
Pain		X	X	Natural enthusiasm of the adolescent			X
Presence of other health problems	X	X	X	Interventions in kinesiology allowing adolescents to be more active and to preserving muscle mass		X	X
Injury/post-surgery problems		X	X	Interventions in kinesiology contributing in increased energy level		X	X
Kinesiology activities being held right after chemotherapy treatment	X	X	X	Interventions in kinesiology helping adolescents return faster to a normal life		X	X
Lack of motivation	X	X	X	General interest of the adolescent in participating in activities			X
Physical deconditioning		X	X	Interventions in kinesiology helping adolescents recover better		X	X
Depression			X	Interventions seen as contributors to reducing treatment side effects		X	
Cancer and treatment side effects	X	X	X	Interventions perceived as helping in weight management		X	
Fatigue/exhaustion	X	X	X				
Negative self-image	X						
Loss of physical capabilities and pride associated to this new physical reality	X						
Loss of hope in their present and future lives	X						
Fear of the judgment from others	X						
Social and academic context of older adolescents		X					
Lack of time		X	X				
Lack of interest		X					
Difficulty accepting new reality	X						
Implementation process	S	A	P	Implementation process	S	A	P
Focus of interventions towards young children	X	X		Interventions being adapted to adolescents’ needs and priorities	X	X	
Adolescents not being placed at the center of decision-making	X			Trustworthy relationship between stakeholders and adolescents	X		
Insistency of some stakeholders for participation of teens		X	X	Stakeholders bringing adolescents together	X	X	
Difficulty understanding the contribution of each stakeholder	X	X	X	Adolescent’s griefs being addressed before beginning of interventions	X		
Lack of coordination between resources	X	X	X	Adolescents being involved in decision-making process	X		
Too much paper forms to fill out			X	Adolescents’ participation being valued by stakeholders	X		

An “x” indicates if the barrier or facilitator was mentioned by a stakeholder, an adolescent and/or a parent. S: stakeholders include oncologists, nurses, physical therapist, occupational therapists, nutritionist, kinesiologists, hospital-based teachers, psychologist. A: adolescents. P: parents.

**Table 4 children-09-01340-t004:** Stakeholders’ perception of the impact of cancer on adolescents.

Physical Changes	Stakeholder Quotations from Focus Groups and Self-Report Questionnaires
Loss of physical capacities	*“It**’**s not everyone who after that [cancer and treatments] is in great shape. Some have complications that are more important than others. Still, fortunately, we have a population that, after all, in general, is relatively well after their treatment.”* (Oncologist, focus group) *“Motivation to participate* *[**in health promotion interventions**]* *is greatest when a person is confident that they have the ability to perform the required movements. Adolescents [who had cancer] know that they are no longer able to do everything the way they used to.”* (Kinesiologist, self-report questionnaires)
Loss of self-esteem/ Body image issues	*“It**’**s a period when they live really difficult things and when their body appearance and self-esteem take a big place.”* (Nurse, focus group) *“We were talking about the relationship to the body. How you perceive your body and all this changes a lot. We can just think of our teenagers on steroids during the induction treatment: they swell a lot and they ask themselves “Will I be able to go back to my pre-treatment weight?” We do what we can to reassure them that they probably will. But the concerns are still present and it**’**s scary for them.”* (Oncologist, focus group)
Psychological/social impact	Stakeholder quotations from focus groups and self-report questionnaires
Many changes at the same time	*“How can I say it… It**’**s all the unknown, everything, everything, everything has changed for them.”* (Hospital-based teacher, focus group)
Feeling that their world is falling apart	*“For them, it**’**s like the real world just collapsed. What they say is “I won**’**t see my friends anymore...””* (Hospital-based teacher, focus group)
Worries for parents/ Strained relationship with parents	*“Parents are afraid for their child. Children are afraid for their parents. And so, at that moment, it becomes a vicious circle of everyone being worried for everyone… Everyone walking on eggshells. And then it becomes a relationship that is tense all the time.”* (Oncologist, focus group)
Lack of understanding from friends	*“This reality would be different if they were in secondary 4 or 5 [grades 10–11]. They [the friends] would understand. But in secondary 1 [grade 7], you don**’**t understand. For older adolescents if someone says “I have cancer”, it**’**s a notion that they know. They**’**d be like ok, we**’**ll see you again when you come back.”* (Oncologist, focus group)
Loss of autonomy	*“For teenagers, around 12 years, maybe 13 years of age, it**’**s the moment when parents are less involved, but you just announced a cancer diagnosis, and they [parents] become very involved. Teenagers want their parents to be involved but at the same time, I do not think they want them to be involved.”* (Oncologist, focus group) *“It**’**s the parents**’* *perception of a sick child: “He needs me.” So the adolescent loses a little of his autonomy sometimes. The disease usually justifies this [overprotection] if it wasn**’**t something that was already present before. But if it was already present, the disease increases their [parents] intensity.”* (Oncologist, focus group)
Diminished decision-making	*“They have lost their authority in their decision-making power.”* (Oncologist, focus group)
Impact on school	*“When we talk to teens about the possibility of having to skip a school year and having to repeat that year, it**’**s like “this cannot be happening to me, it**’**s ONE year”, it**’**s terrible for them.”* (Hospital-based teacher, focus group)
Challenges and grief of adolescents in oncology	Stakeholder quotations from focus groups and self-report questionnaires
Missing themselves as they were before cancer	*“[Teens are like] “I miss myself as I was before, I miss my old body, I miss my friends and I miss the interests I used to have, that were the same as my friends. Now, I don**’**t care about shoes and color of shoes and all that.””* (Psychologist, focus group) *“There is all this grief: grief of me, of how I [the adolescent] was before. And that**’**s where all the self-esteem comes in. But it**’**s really like: “I [the adolescent] don**’**t know if I will be able to go back to how I was before* *[the* *cancer diagnosis**]**, I would like things to go back as they were before.””* (Psychologist, focus group)
Grief for activities	*“I am thinking of this teen who was really grieving not being able to go on her snowboard at the first snow because she had a big pneumopathy. You can present to her any other alternative activity, but she will not do it.”* (Physiotherapist, focus group) *“I find that there is the challenge to mobilize them to move forward.* *[**…] They [adolescents**]* *can say, among other things. “Ok, I want to go skiing”, but they want to go from “I**’**ve been lying down for 3 months” to “I**’**m going skiing tomorrow morning.” It**’**s the in-between that must be done in rehabilitation, that they do not want, because they would like to already be there, it is the in between, the path to get there, that is difficult.”* (Occupational therapist, focus group)
People perceiving them only as a teenager who has cancer	*“I [the adolescent] am more than a young person with cancer. I want to be recognized for something else. But that**’**s because when they go to LEUCAN [a non-profit foundation that supports cancer-stricken children and their families] activities, they say, “I**’**m going to have to talk about cancer”. I tell them: No, you don**’**t have to talk about cancer. They are together, all young people, and they finally end up talking about it, but that**’**s not what triggers it: They talk about all kinds of things. And then, yes, at some point, they ask others: “So, what is it [cancer type] that you have?”* (Psychologist, focus group)
Loss of friends/ No sense of belonging	*“I [the adolescent] miss myself in a gang, with my group, which I am losing” And for many, it got worse with the pandemic.”* (Psychologist, focus group) *“We have a girl right now who is in secondary 1 [grade 7]. I was talking to her the other day about her friends at school. She was like "what friends?" And I was like, well you have friends at school. But she was like no… because her friends, they don**’**t understand what she**’**s going through.”* (Hospital-based teacher, focus group)
Things will never be the same again	*“I talk to them a lot about [the fact that] it will never be like before. It**’**s hard to hear that. I say the same thing to parents. It**’**s very hard, what I tell them, and I tell them that I know it**’**s hard to hear, but that**’**s the reality. There will never be a “like before”, because you are living this experience. But I also tell them that there are going to be beautiful things coming anyway.”* (Psychologist, focus group)

**Table 5 children-09-01340-t005:** Recommendations to adapt oncology lifestyle promotion interventions for adolescents.

Recommendations	Quotations from Stakeholders, Adolescents or Parents
Build trust in the relationship	*“Trust is good. […] But complicity is more than that: it**’s making jokes and discovering what they are passionate about and what thrills them.”* (Hospital-based teacher, focus group) *“If they see you as a person with whom they feel like spending time with and not a person who comes to tell them what to do, I think that it would already be great and a huge plus.”* (Oncologist, focus group) *“I believe that being able to generate the necessary trust for them to tell us about their new feelings, fears, preferences and motivations can serve as […] a way to help them achieve their new goals or manage their new reality in a better way.”* (Kinesiologist, self-report questionnaire)
Keep in mind treatments may impact mood	*“She [an adolescent] had an antisocial spike at one point. It was quite impressive… The Decadron really affected her… It was really bad…”* (Hospital-based teacher, focus group)
Balance constancy and relentlessness	*“It is important for them [the adolescents] to be able to have the choice of participating or not in something that is optional. In fact, I think there**’s a balance between relentlessness, […] to go back constantly and bother them, and to let them know that we**’re here for them.”* (Oncologist, focus group)
Give adolescents the right to change their mind	*“There are many [adolescents] who refuse [to participate] from the beginning. We will go back later, we will give them time to think about it, and at one point they accept.”* (Hospital-based teacher, focus group)
Make sure the adolescent perceives the relevance of the intervention	*“We can just present them what we do. […] I tell them: “I am your physiotherapist; I can do your treatments. I can see you or not see you, it is you who decides. I can do this, this, and that for you. I can teach you things. When you**’re ready, tell me, and I will come.” It can take a month or even a few months before I do anything. […] My needs are not the same as their needs. […] I just wait for them to need me to go help them.”* (Physiotherapist, focus group)
Help adolescents regain their autonomy	*“What I like a lot is to make them autonomous, to take charge of their life, their health, to give them the tools and the possibility to do it.”* (Occupational therapist, focus group) *“At some point, he**’s going to need me, I know. He**’s going to need me to walk. He**’s going to need me to get up and at that point, he will know that someone is there to help him. It**’s just... It**’s like you said: planting a little seed. Then, they [the adolescents] know that the person is there when they need them.”* (Physiotherapist, focus group)
Have a trusted stakeholder make the first approach	*“They [the adolescents] tend to say no because they don**’t know exactly what it is [the intervention], but if it comes from someone they trust, I think it can help them make their decision.”* (Oncologist, focus group)
Favor non-moralizing teachings	*“[We have to] be careful not to present things as being moralizing, in the sense that: “WE will explain to you how you should eat.””* (Hospital-based teacher, focus group)
Avoid using long-term health as a motivator	*“You do not realize at 12, 13, 14 years old, that you must take care of your body, in the sense that, for them, it is the present moment [that matters].”* (Hospital-based teacher, focus group) *“Lifestyle habits are part of a long-term concept, which is probably less of a priority for adolescents in the short term. Since it is less "relevant" to them in the short term, they are less involved.”* (Nutritionist, self-report questionnaire) *“They need to find an external motivation to move. Health is too an abstract concept for it to be a reason for them to be active (even if they are sick).”* (Kinesiologist, self-report questionnaire)
Build objectives with the adolescent	*“Teenagers all have a previous life. […] [I ask them:] “Where are you starting from? And what is your goal? I**’m going to work with you, I**’m going to take you wherever you want to go.” […] Maybe it would be to target an objective directly with the adolescent. It**’s having the health professional on board with the adolescent. It**’s also about aligning our goals and making them evolve together.”* (Physiotherapist, focus group) *“I believe that starting from their objectives can be an important key. In order to discover this objective, it is necessary to develop a relationship of trust. A process that can take time, but it is worth it and requires commitment and consistency in the approach.”* (Kinesiologist, self-report questionnaire)
Celebrate victories	*“It**’s also about measuring [objectives] and celebrating victories. […] It**’s to make them realize that they spent 20 min less per day on their tablets. […] When you make them realize that they went to sit in their chair 3 times a day and you try to integrate a little bit of physical activity.”* (Physiotherapist, focus group)
Modernize the approach/Adapt interventions to adolescent’s limited attention span	*“We have to modernize a bit the way we communicate with them. I think that the intervention is important, but the way we get the message across is important too. [..] Maybe it**’s not a one-hour meeting with a nutritionist, but maybe it**’s a YouTube clip of I don**’t know how long, maybe a minute? We know on Tik Tok, it**’s 15 s. That**’s the attention that people have. And it**’s the same for adults.”* (Oncologist, focus group)
Make interventions attractive	*“It would be interesting to optimize the intervention to make it more attractive and more stimulating for adolescents. To involve them more, to develop virtual content, to make teenagers interact together in a group.”* (Nutritionist, self-report questionnaires).
Develop adolescents’ physical literacy/Develop their confidence and motivation	*“Physical literacy is defined as “the motivation, confidence, physical competence, knowledge and understanding that a person possesses and that allows them to take charge of their commitment to physical activity for the rest of their life." […] “To summarize, I think their [the adolescents] specific needs are to develop their confidence and motivation in a context of lifestyle habits.”* (Kinesiologist, self-report questionnaires)
Ask adolescents their preferences before developing an intervention	*“The other thing I would say, actually, is to have teenagers either in treatment or who have finished their treatment with us in this discussion. It could really be relevant because, you know, we have ideas, but maybe they**’re going to be like “No, not at all, you**’re**’stupid**’ to think like that.”* (Oncologist, focus group)
Bring adolescents together/ Develop specific activities for them	*I think we should bring them [adolescents] together and develop specific activities for them that are not moralistic.* (Nutritionist, self-report questionnaire)
Have a space dedicated for adolescents in oncology	*“A few years ago, we had thought about this. You know: “how can we make it easier for adolescents to spend their days here?” And that**’s when the idea of having a room for adolescents emerged.”* (Psychologist, focus group) *“We could ask LEUCAN [a non-profit foundation that supports cancer-stricken children and their families] to organize something for our teenagers… They don**’t need something extraordinary. We won**’t ask for a loft with non-alcoholic champagne or whatever. But we already have a physical space here, we already have a TV, you do not add the cable, you just make sure to have Netflix, Disney and Prime or whatever. That would make their waiting time here already much easier.”* (Oncologist, focus group)
Address the griefs before initiating activities	*“I think there are a lot of griefs to deal with before you even start an activity. Finding a way to include friends could be really positive, I think. Show them that they can still do a lot of things despite their illness. Maybe discovering other sports or activities that they can**’t compare to "before the disease."”* (Kinesiologist, self-report questionnaire)
Respect adolescents’ schedules	*“I had a lot of sleep problems with the chemotherapy treatments and I would sleep here in the morning at the hospital. And the ladies from the VIE project respected that I was sleeping. They didn**’t wake me up, contrary to other people in other projects. They understood that if I was sleeping, it was because I needed to sleep.”* (Male, 16 years old at diagnosis)
Plan interventions between medical appointments	*"For sure that it**’s good between appointments. Sometimes they**’d come and get me and we**’d play badminton. It helped to pass time between appointments because sometimes it**’s weird, they give you an appointment at 12:30 pm and the other one is at like 3:45 pm."* (Male, 15 years old at diagnosis).
Consider treatment side effects when scheduling visits	*"Sometimes they would come see me like after chemo and I was feeling so sick, I know I wouldn**’t have been able bend over and all that. That**’s when it didn**’t work."* (Male, 15 years old at diagnosis). *“It wasn**’t that bad... Well, maybe when we were doing our training sessions, the timing wasn**’t always right. Sometimes they would come either while I was getting chemo, I had my pole and all, or afterwards. It wasn**’t always the [best timing]. Let**’s say I would have done it before [the chemo] maybe.”* (Female, 15 years old at diagnosis)
Personalize and adapt the interventions	*"It helped me come up with a home exercise plan that was more tailored to what I was able to do. Like, it made me get better, but it didn**’t get to the point where it hurt me..."* (Male, 15 years old at diagnosis). *“I had several restrictions because everyone is different and they all have different restrictions and they [stakeholders] adapted to what I could and couldn**’t do when I could and when I couldn**’t do it. They were very flexible.”* (Female, 13 years old at diagnosis)
Offer remote interventions	*“You ask yourself where do you put your energy to save her [adolescent daughter with cancer]? But the fact remains that there are 3 others at home... […]. It complicates the schedule I would say... The schedule management... […] You have the VIE project that comes along, and they tell you: look we help you with nutrition, we help you with physical activity, […] And, even better, activities were being held remotely. With COVID [-19 pandemic], we really appreciated the remote activities...”* (Father of female, 13 years old at diagnosis) *“Well, personally, I found that we had more remote services with the pandemic, than when we did before... You know, when we would come here [at the hospital], we**’d come once a month. Then, we had weekly exercises... You know, for me the pandemic was kind of a lifesaver…”* (Father of male, 12 years old at diagnosis)
Adapt education content to older adolescents	*"Make people understand: “Look, I**’m making you do this, it**’s going to help you with that.” [...] I think that could have helped me. Maybe less so for children like 4–5 years of age. But I was 18 at the time. I think it could have helped me understand why I was doing it [exercise], not just the way it should be done. [...] I think to help young people who are more like teenagers, 13–14 years old, to understand: “Ok, you know, I**’m not here to piss you off or to make you sweat there, I**’m here to help you, look it**’s going to do that [for you].””* (Female, 17 years old at diagnosis)
Remember that the adolescent may be having a difficult day	*“During treatments, I often felt unwell so I... You feel like you**’re weak and your whole-body hurts, so it**’s hard to say: “Okay, this morning I**’m going to get up and I**’m going to go to the gym with the physiotherapists...””* (Female patient, 17 years old at diagnosis).
Offer follow-up in the longer term if needed	*“My expectation with respect to the rest of the project, well if you can continue to help us... You know when the physio will be finished and all that… For sure we would appreciate... to find out how we can bring [daughter**’s name] back to her 100%. And I know this is part of a research project […], but if you see in your observations that there are certain problems that you can help with, then... offer that too. You can tell the hospital that you followed a participant and that there might be a need for that [specific] long-term [health] care for this patient…”* (Father of male, 12 years old at diagnosis)

## Data Availability

Not applicable.

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
