# Peer review of "Needs, Barriers and Facilitators of Adolescents Participating in a Lifestyle Promotion Program in Oncology: Stakeholders, Adolescents and Parents’ Perspective"

_children, 2022, doi:10.3390/children9091340_

Round 1

Reviewer 1 Report

This is a qualitative analysis involving data collected from adolescents with cancer, their parents, and oncology stakeholder’s to elicit their perspectives on participating in a lifestyle promotion program, and the associated barriers, facilitators and specific needs. Overall, the paper is well written and interesting, although does lack some detail particularly in the presentation of the methods.

ABSTRACT

1.       The abstract opens with mention of the effects of treatment for paediatric cancer, causing confusion about the study’s sample i.e. are these adolescent aged survivors of paediatric cancer, or young people diagnosed during adolescence? Perhaps rephrase as appropriate to the relevant sample.

2.       It would be useful to add basic information about the target sample to the abstract e.g. mean age, time since diagnosis or other relevant information available

3.       Currently, facilitators and barriers are grouped, with no acknowledgement of potential similarities/differences in the participant groups (e.g. adolescent’s vs parent’s vs stakeholder’s reported or perceived barriers/facilitators)

INTRODUCTION

4.       Page 2, paragraph 2: Overall, the introduction offers a succinct yet comprehensive summary of the long-term health issues faced by adolescents with cancer and the importance of promoting healthy lifestyles. To further strengthen the rationale for this study, further consideration might be given to incorporating i) targeting lifestyle/health promotion during cancer treatment (say as opposed to post-treatment or long-term survivorship), and ii) the gaps that the other existing interventions which include adolescents have, and the resulting need for further programs (are these interventions age-appropriate? not tailored for each of the age groups, since being offered to all ages?)

5.       Page 2, paragraph 3: The premise of the current study is that the team’s existing evaluation of a nutritional intervention resulted in low uptake among adolescents, and the authors suggest the lower engagement may be due to the need to adapt the intervention to adolescents’ preferences and needs. Is there a basis for this suggestion? Suggest substantiating this hypothesis, especially since it guides the secondary objective of the study.

METHODS

6.    Page 3, paragraph 1: Whilst I appreciate that there have been other publications describing the VIE, it would be great to include a brief summary here for completeness (i.e. it’s mentioned later that participants must have participated in the VIE study for 1.5 years, but the longitudinal design, methods etc aren’t obvious)

7.    Page 3, paragraph 3: What was the rationale for only including patients who had been treated with chemotherapy or radiotherapy (and not surgery, transplant etc)? It’s also unclear if there were any exclusion around the time since diagnosis. Were these patients still on treatment?

8.    Page 3, paragraph 5-6: What was the purpose of having a stakeholder focus group and survey? Whilst the content of the surveys was generally clear, it’s not clear how this differed from the data collected in the focus groups. Did the same stakeholders complete focus groups and surveys? What questions were asked; measures used (if relevant)? Presumably the survey questions were open-ended in order to facilitate thematic analysis of these data?

RESULTS

9.       Table 3 is a nice summary of the key findings, and endorsement by each participant group. Throughout the results, suggest carefully revising to ensure appropriate reporting of results by each participant group e.g. avoiding “adolescents treated for cancer also have a need…” to “stakeholders perceived that adolescents treated for cancer also have a need…”, and in the presentation of adolescent and parent results/themes collectively, where in fact there may be important differences. I think this is important to more accurately reflect the opinions of adolescents, and the perceptions of parents and stakeholders about adolescents

10.   Page 10: the section on the impact of cancer on adolescents is insightful, but also strays from the two articulated objectives. As noted above, it might help to more clearly describe the measures/questions to avoid ‘surprises’ in the results, and/or revise the objectives as appropriate

DISCUSSION

11.   Page 16: Given the suggested novelty of the finding regarding adolescents wanting enough time to decide about participation in an intervention, the authors may consider further reflecting on this point and the potential implications

12.   Page 17, paragraph beginning “most studies evaluating…” does not add significant value to the discussion in my opinion and could be omitted

13.   Page 17, paragraph starting “stakeholders participating in…” also does not add much value, although does relate to the paragraph at the top of the page on other physical barriers to participation noted by parents/adolescents and could be merged, or otherwise omitted

14.   The discussion overall summarizes many of the key findings well, and engages with the broader literature on this topic. However, it could be strengthened by further reflection on the implications of these findings for future research, development of interventions, and/or the delivery of services and lifestyle programs for this population

15.   I think another strength worth noting is that around half of parents were fathers, who are often underrepresented

Author Response

Comments and Suggestions for Authors

Reviewer #1

This is a qualitative analysis involving data collected from adolescents with cancer, their parents, and oncology stakeholder’s to elicit their perspectives on participating in a lifestyle promotion program, and the associated barriers, facilitators and specific needs. Overall, the paper is well written and interesting, although does lack some detail particularly in the presentation of the methods.

ABSTRACT

Comment 1. The abstract opens with mention of the effects of treatment for paediatric cancer, causing confusion about the study’s sample i.e. are these adolescent aged survivors of paediatric cancer, or young people diagnosed during adolescence? Perhaps rephrase as appropriate to the relevant sample.

Response: We thank the reviewer for all his/her recommendations. The sentence has been modified:

Line 26, previous version:

Treatments for pediatric cancer can cause debilitating side effects in the short- and long-term such as nausea and malnutrition but also cardiometabolic disturbances.

Line 26, new version:

Treatments for adolescent cancer can cause debilitating side effects in the short- and long-term such as nausea and malnutrition but also cardiometabolic disturbances.

Comment 2. It would be useful to add basic information about the target sample to the abstract e.g. mean age, time since diagnosis or other relevant information available

Response: According to the Reviewer’s comment, a sentence was added in the Abstract. We have also modified the text to include the number of adolescent participants and the number of parent participants in the abstract. We have also modified the text to include the exact number of parents and adolescents who had participated in interviews.

Line 31 previous version:

Interviews were held with adolescents treated for cancer and parents (n=13), focus groups were conducted with stakeholders (n=12) working in oncology and self-report questionnaires were sent to stakeholders (n=6) involved in a health promotion intervention. Verbatim and responses to questionnaires were coded and analyzed using qualitative methods.

Text, new version:

Interviews were held with adolescents treated for cancer (n=9) and parents (n=6), focus groups were conducted with stakeholders working in oncology (n=12) and self-report questionnaires were sent to stakeholders involved in a health promotion intervention (n=6). At the time of interview, mean age of adolescent participants (40% female) was 17.0 ± 1.9 years (mean age at diagnosis: 14.6 ± 1.6 years).Verbatim and responses to questionnaires were coded and analyzed using qualitative methods.

Comment 3. Currently, facilitators and barriers are grouped, with no acknowledgement of potential similarities/differences in the participant groups (e.g. adolescent’s vs parent’s vs stakeholder’s reported or perceived barriers/facilitators)

Response: We thank the reviewer for pointing this out to us.

We have added the information to the abstract:

Section of the abstract starting at line 35, previous version:

Results show that adolescents treated for cancer need to access activities adapted to their age, to communicate with peers going through a similar experience, and to preserve their schooling and friendships. Barriers to intervention include lack of motivation, schedule conflicts, fatigue and treatment side effects. Facilitators comprise trust in stakeholders’ expertise, personalized approaches, scheduling flexibility and respect of the adolescents’ limits by the stakeholders. Recommendations to adapt a health promotion intervention to adolescents’ needs comprise building trust in the relationship, favoring non-moralizing teachings, adapting interventions to adolescents’ limited attention span and avoiding the use of long-term health benefits as a motivator. 

Section of the abstract starting at line 36, new version:

Stakeholder stated that adolescents with cancer need to access activities adapted to their age, to communicate with peers going through a similar experience, and to preserve their schooling and friendships. Barriers to intervention reported by adolescents, parents and stakeholders include lack of motivation, schedule conflicts, fatigue and treatment side effects. Some of the barriers mentioned exclusively by adolescents and parents include pain, post-surgery problems, school, physical deconditioning, and lack of time. Facilitators mentioned by adolescents and parents comprise trust in stakeholders’ expertise, personalized approaches, scheduling flexibility. Stakeholders recommended to build trust in the relationship, favor non-moralizing teachings, adapt interventions to adolescents’ limited attention span and avoid the use of long-term health benefits as a motivator. 

INTRODUCTION

Comment 4. Page 2, paragraph 2: Overall, the introduction offers a succinct yet comprehensive summary of the long-term health issues faced by adolescents with cancer and the importance of promoting healthy lifestyles. To further strengthen the rationale for this study, further consideration might be given to incorporating i) targeting lifestyle/health promotion during cancer treatment (say as opposed to post-treatment or long-term survivorship), and ii) the gaps that the other existing interventions which include adolescents have, and the resulting need for further programs (are these interventions age-appropriate? not tailored for each of the age groups, since being offered to all ages?)

Response: We thank the Reviewer for his/her insightful comment. The following text was added to the Introduction section between paragraphs 2 and 3:

Early changes in lifestyle habits in children and adolescents with cancer may occur early after the initiation of treatments (ref. Stern, 2013, Fleming, 2015, Williams, 2015). Despite this knowledge, health promotion interventions in oncology were mainly held after the end of treatment period or during survivorship (Cohen, 2016, Devine, 2018). Health promotion interventions that begin early after cancer diagnosis may be key element to establish healthy lifestyle habits and prevent cancer late effects.

These references were added:

Stern, M.; Lamanna, J.; Russell, C.; Ewing, L.; Thompson, A.; Trapp, S.; Bitsko, M.; Mazzeo, S. Adaptation of an obesity intervention program for pediatric cancer survivors (NOURISH-T). Clin. Pract. Pediatr. Psychol. 2013, 1, 264–275

Fleming, C.A.; Cohen, J.; Murphy, A.; Wakefield, C.E.; Cohn, R.J.; Naumann, F.L. Parent feeding interactions and practices during childhood cancer treatment. A qualitative investigation. Appetite 2015, 89, 219–225.

Williams, L.K.; McCarthy, M.C. Parent perceptions of managing child behavioural side-effects of cancer treatment: A qualitative study. Child. Care Health Dev. 2015, 41, 611–619.

Cohen JE, Wakefield CE, Cohn RJ. Nutritional interventions for survivors of childhood cancer. Cochrane Database of Systematic Reviews 2016, Issue 8. Art. No.: CD009678. DOI: 10.1002/14651858.CD009678.pub2.

Devine, K. A., Viola, A. S., Coups, E. J., & Wu, Y. P. (2018). Digital Health Interventions for Adolescent and Young Adult Cancer Survivors. JCO clinical cancer informatics, 2, 1–15. https://doi.org/10.1200/CCI.17.00138

Comment 5. Page 2, paragraph 3: The premise of the current study is that the team’s existing evaluation of a nutritional intervention resulted in low uptake among adolescents, and the authors suggest the lower engagement may be due to the need to adapt the intervention to adolescents’ preferences and needs. Is there a basis for this suggestion? Suggest substantiating this hypothesis, especially since it guides the secondary objective of the study.

Response: We thank the Reviewer for pointing out this important comment. These modifications have been performed:

Line 88, previous version:

The reasons explaining the lower engagement of adolescents are not clear, but we suggested that the design of the family-based intervention was not adapted to adolescents’ preferences and needs.

Text, new version:

The reasons explaining the lower engagement of adolescents are not clear, but we have suggested that it could partly be explained to the family-based design of the intervention which may not be adapted to adolescents’ preferences and needs. Outside the field of oncology, many health prevention and promotion programs designed exclusively for adolescents were school-based as it was shown to be an effective strategy for the adolescent population (Ref Lassi, 2015). In oncology, adolescents with cancer have reported different problems and concerns than younger children (Ref Lewandowska, 2021), but whether this is also true in the context of health promotion interventions in unknown.

References to be added:

Lassi, Z. S., Salam, R. A., Das, J. K., Wazny, K., & Bhutta, Z. A. (2015). An unfinished agenda on adolescent health: Opportunities for interventions. Seminars in perinatology39(5), 353–360. https://doi.org/10.1053/j.semperi.2015.06.005

Lewandowska, A., Zych, B., Papp, K., Zrubcová, D., Kadučáková, H., Šupínová, M., Apay, S. E., & Nagórska, M. (2021). Problems, Stressors and Needs of Children and Adolescents with Cancer. Children (Basel, Switzerland)8(12), 1173. https://doi.org/10.3390/children8121173 

METHODS

Comment 6. Page 3, paragraph 1: Whilst I appreciate that there have been other publications describing the VIE, it would be great to include a brief summary here for completeness (i.e. it’s mentioned later that participants must have participated in the VIE study for 1.5 years, but the longitudinal design, methods etc aren’t obvious)

Response: According to the Reviewer’s comment, we have added the following text in the Methods section:

Paragraph starting at line 99, previous version:

This research is part of the VIE (Valorization, Implication, Education) study (web link: VIE Study) conducted at the CHU Sainte-Justine (CHUSJ) in Montreal (Quebec, Canada). The proposed integrative and personalized program informs newly diagnosed patients, their families, and healthcare professionals about the benefits of adopting early healthy lifestyles, particularly a balanced diet, physical activity (PA), and stress management. The design of the interventions has been described in detail [54,57,58]. The current work consists in a qualitative study conducted with a subgroup of the VIE study adolescent participants, in addition to their parents, and stakeholders working in oncology at CHUSJ (health care and research personnel). Participants were recruited from January 2018 until December 2021. Data were collected using focus groups, questionnaires, and interviews. This study was approved by the Ethics Review Committee of the CHUSJ (#2021-3129 and #2021-3278).

Paragraph, new version:

This research is part of the VIE (Valorization, Implication, Education) study (web link: VIE Study), a non-randomized controlled feasibility study conducted at the CHU Sainte-Justine (CHUSJ) in Montreal (Quebec, Canada). The proposed integrative and personalized program informs newly diagnosed patients and their families about the benefits of adopting early healthy lifestyles, particularly a balanced diet, physical activity (PA), and stress management to improve quality of life and prevent the incidence of late effects. The design of the interventions has been described in detail [54,57,58]. Briefly, participants 21 years old or younger at diagnosis; treated with chemotherapy or radiotherapy and able to give informed consent (by parents or legal guardians) were invited to participate. Patients were excluded if they were not being treated with chemotherapy and/or radiotherapy. Eligible participants were enrolled between the fourth and twelve weeks after cancer diagnosis. At recruitment, participants underwent an initial evaluation in nutrition, PA and psychological health. Follow-up visits were planned every month by alternating nutrition/PA consults. In parallel, the psychosocial support program was deployed in vulnerable families. Comprehensive nutrition, PA and psychological health evaluations were performed one year after the beginning of the intervention and after the end of cancer treatment. The control group was recruited from patients who had completed their treatment using the same criteria as for the protocol intervention group. No intervention was offered to the control group participants other than standard clinical follow-ups. The current work consists in a qualitative study conducted with a subgroup of the VIE study adolescent participants, in addition to their parents, and stakeholders working in oncology at CHUSJ (health care and research personnel). Participants were recruited from January 2018 until December 2021. Data were collected using focus groups, questionnaires, and interviews. This study was approved by the Ethics Review Committee of the CHUSJ (#2021-3129 and #2021-3278).

Comment 7.  Page 3, paragraph 3: What was the rationale for only including patients who had been treated with chemotherapy or radiotherapy (and not surgery, transplant etc)? It’s also unclear if there were any exclusion around the time since diagnosis. Were these patients still on treatment?

Response: In the VIE study, patient who did not undergo chemotherapy or radiotherapy were not included because the study main objective was to test the feasibility of a multi-component intervention during pediatric cancer treatment and initiated early after diagnosis. Patients who are not treated with chemotherapy or surgery are often not hospitalized or followed for sufficient time to justify participation in the VIE study. Also, they do not experience acute and long-term side effects of treatments.

Hematopoietic stem cell transplant was not an exclusion criteria. However, newly diagnosed patients rarely receive this treatment from the start and thus are usually treatment with chemotherapy and/or surgery at first.

The inclusion criteria for the reported qualitative study were the same as for the VIE study, with the additional criteria to having participated in the VIE study for a minimum of 1.5 years. This is because it was necessary for them to have participated in the study for a long enough time to be able to express their perceptions about the interventions.

All adolescent participants in the current study had ended their treatment at the time of the study interviews. To clarify this point, we added text in the Method section – 2.3.3  Interviews with adolescents and parents :

Paragraph starting at line 149, previous version:

All adolescents and parents participating in the VIE study were invited to meet the interviewer for a face-to-face interview. Two types of semi-structured interviews were conducted: barriers and facilitators interviews and exit interviews. We established a priori that participants could not take part in both types of interviews. A total of 31 in person interviews (n=31), conducted individually or as an adolescent-parent dyad, were audio recorded. For the current study, only interviews conducted with adolescents and parents of adolescents were considered (n=12), including 8 individual interviews about barriers and facilitators and 4 exit interviews. Interview duration ranged from 8 to 32 minutes for a total of 194 minutes of verbatim. To reduce bias, all interviews were conducted by the same person (C.D.) who is an occupational therapist and doctorate candidate.

Paragraph, new version:

All adolescents and parents participating in the VIE study were invited to meet the interviewer for a face-to-face interview. Two types of semi-structured interviews were conducted: barriers and facilitators interviews and exit interviews. We established a priori that participants could not take part in both types of interviews. A total of 31 in person interviews (n=31), conducted individually or as an adolescent-parent dyad, were audio recorded. For the current study, only interviews conducted with adolescents and parents of adolescents were considered (n=12), including 8 individual interviews about barriers and facilitators and 4 exit interviews. At the time interviews were conducted, cancer treatments were completed for all adolescent participants. Interview duration ranged from 8 to 32 minutes for a total of 194 minutes of verbatim. To reduce bias, all interviews were conducted by the same person (C.D.) who is an occupational therapist and doctorate candidate.

Comment 8.  Page 3, paragraph 5-6: What was the purpose of having a stakeholder focus group and survey? Whilst the content of the surveys was generally clear, it’s not clear how this differed from the data collected in the focus groups. Did the same stakeholders complete focus groups and surveys? What questions were asked; measures used (if relevant)? Presumably the survey questions were open-ended in order to facilitate thematic analysis of these data?

Response: We agree with the reviewer that the description provided in the Method section is confusing. There were 2 different categories of stakeholders who participated in our study: 1) healthcare professionals and hospital-based teachers working with adolescents in oncology at CHUSJ who participated in the focus groups and 2) research health personnel who delivered the VIE study intervention, comprising of research kinesiologists and nutritionists who were eligible to answer the self-report questionnaire. Therefore, the stakeholders who participated in the focus group were not the same as those who completed the self-report questionnaire.

All survey questions were open-ended. Specifically, the questions were:

  • What are the specific needs of adolescents in an oncology setting?
  • What can negatively affect the participation of adolescents in lifestyle interventions?
  • What could be done to optimize the participation of adolescents in lifestyle interventions?
  • When do you think is the best time to initiate a lifestyle intervention for adolescents in oncology?

In the present article, we did not develop on the fourth question as it deviated from the aims.

According to the Reviewer’s comments, these changes have been performed in the Methods section – 2.2.1 Stakeholders:

Paragraph starting at line 113, previous version:

Stakeholders were eligible for participation in focus groups if they were: 1) a healthcare professional or a hospital-based teacher working with children and adolescents in oncology at CHUSJ and 2) exposed to the VIE study (health care personnel). Stakeholders were eligible to answer the written questionnaires if they: 1) were a member of the VIE research team, 2) had been working directly with participants in the nutrition or PA components study (research personnel). 

Paragraph, new version:

Two categories of stakeholders were included in the study: 1) healthcare professionals and hospital-based teachers working with adolescents in oncology at CHUSJ who participated in the focus groups and 2) research health personnel who delivered the VIE study intervention, comprising of research kinesiologists and nutritionists who were eligible to answer the self-report questionnaire. Stakeholders were eligible for participation in focus groups if they were: 1) a healthcare professional or a hospital-based teacher working with children and adolescents in oncology at CHUSJ and 2) exposed to the VIE study (health care personnel). Stakeholders were eligible to answer the self-report questionnaire if they: 1) were a member of the VIE research team, 2) had delivered the nutrition (nutritionists) or PA (kinesiologists) interventions. 

RESULTS

Comment 9. Table 3 is a nice summary of the key findings, and endorsement by each participant group. Throughout the results, suggest carefully revising to ensure appropriate reporting of results by each participant group e.g. avoiding “adolescents treated for cancer also have a need…” to “stakeholders perceived that adolescents treated for cancer also have a need…”, and in the presentation of adolescent and parent results/themes collectively, where in fact there may be important differences. I think this is important to more accurately reflect the opinions of adolescents, and the perceptions of parents and stakeholders about adolescents

Response: We thank the Reviewer for his/her valuable comment. We have performed the following changes to address it.

Section 3.3 starting at line 218, previous version:

Stakeholders agreed on the fact that adolescents should be considered as a distinct group with regards to healthcare services as their motivations and needs are different from those of children and adults. One of the recurrent themes was adolescents’ need for autonomy and freedom, the need to break free and test limits. Their need to have a sense of control and power was also mentioned. Stakeholders raised the importance of including adolescents in decision making, including objective settings, instead of im-posing on them the priorities of the study and stakeholders.

Specific to adolescents in oncology, stakeholders mentioned adolescents’ need to have access to services and interventions adapted to their age, to connect with other adolescents experiencing a similar situation, to regain a positive body image and to maintain their physical capacities. Adolescents treated for cancer also have a need for independence, for their personal space, for time without their parents, to express their feelings and anger, to discover new enjoyable activities adapted to their new reality and medical restrictions. A comprehensive description of the needs of adolescents in oncology identified in our study is presented in Table 2, including stakeholders’ quotations.

Section 3.3 new version:

Stakeholders agreed on the fact that adolescents should be considered as a distinct group with regards to healthcare services as their motivations and needs are different from those of children and adults. One of the recurrent topics mentioned by stakeholders was adolescents’ need for autonomy and freedom, and the need to test limits. One oncologist stated that adolescents need to have a sense of control and to make their own decisions. Stakeholders raised the importance of including adolescents in decision making, including objective settings, instead of imposing on them the priorities of the study and stakeholders.

Specific to adolescents in oncology, many stakeholders mentioned that adolescents need to have access to services and interventions adapted to their age, to connect with other adolescents experiencing a similar situation, to regain a positive body image and to maintain their physical capacities. Stakeholders, including healthcare professionals, research personnel and hospital-based teachers expressed that adolescents treated for cancer have a need for independence, for their personal space, for time without their parents, to express their feelings and anger, to discover new enjoyable activities adapted to their new reality and medical restrictions. A comprehensive description of the perception of stakeholders regarding the needs of adolescents in oncology expressed during focus groups or in self-report questionnaires is presented in Table 2, including stakeholders’ quotations.

Also, title of Table 2 has been modified:

Line 233, previous version:

Table 2. Specific needs of adolescents in oncology

Title of Table 2, new version:

Table 2. Stakeholders’ perception of the specific needs of adolescents in oncology

Comment 10. Page 10: the section on the impact of cancer on adolescents is insightful, but also strays from the two articulated objectives. As noted above, it might help to more clearly describe the measures/questions to avoid ‘surprises’ in the results, and/or revise the objectives as appropriate

Response: We thank the Reviewer for his/her recommendation. This text was added in the Results section of the manuscript:

Section 3.5- Impact of cancer in adolescents, Line 257, previous version:

Stakeholders participating in this study brought insights with regards to the multi-level impact of having a cancer diagnosis during adolescence. They mentioned the physical consequences of cancer that often lead to a poor body image, but they also referred to the loss of physical capacities which leads to the loss of self-esteem. Stakeholders also reported the psychological and social consequences of having cancer during adolescence. They indicated that adolescents face many changes at the same time and that their world appears to be falling apart. Adolescents being treated for cancer often have strained relationships with their parents and perceive a lack of understanding from their friends. The main consequences of cancer mentioned by stakeholders during focus groups and in written questionnaires are highlighted in Table 4.

Section 3.5- Impact of cancer in adolescents, new version:

Although this was not a specific study objective nor was it a precise question asked during focus groups or included in the self-report questionnaire, several additional insightful discussion topics raised during focus groups and were cited in the self-report questionnaire. Most of them were related to the multi-level impact of having a cancer diagnosis during adolescence. Stakeholders mentioned the physical consequences of cancer that often lead to a poor body image, but they also referred to the loss of physical capacities which leads to the loss of self-esteem. Stakeholders also reported the psychological and social consequences of having cancer during adolescence. They indicated that adolescents face many changes at the same time and that their world appears to be falling apart. Stakeholders also mentioned that adolescents being treated for cancer often have strained relationships with their parents and perceive a lack of understanding from their friends. Arising results pertaining to consequences of cancer mentioned by stakeholders during focus groups and in self-report questionnaires are highlighted in Table 4.

We will also change the title of Table 4:

Line 266, previous version:

Table 4. Impact of cancer on adolescents

Title of Table 4, new version:

Table 4. Stakeholders’ perception of the impact of cancer on adolescents

DISCUSSION

Comment 11. Page 16: Given the suggested novelty of the finding regarding adolescents wanting enough time to decide about participation in an intervention, the authors may consider further reflecting on this point and the potential implications

Response: We thank the Reviewer for his/her comment. We have added this text in the Discussion section:

Text starting at line 322, previous version:

Another emergent theme pertained to the importance of allowing adolescents enough time to decide whether they want to participate or not in a healthy lifestyle intervention. To our knowledge, it is the first time this concept has been reported in the context of adolescent oncology.

Text, new version:

Another emergent theme pertained to the importance of allowing adolescents enough time to decide whether they want to participate or not in a healthy lifestyle intervention. To our knowledge, it is the first time this concept has been reported in the context of adolescent oncology. In view of this finding and in consideration for the deployment of future health promotion interventions specific for adolescents, we suggest not to overly limit the recruitment period to allow them time to decide whether or not they want to participate. Offering the possibility to enrol in the intervention at different times during the treatment trajectory could also meet this need.

Comment 12. Page 17, paragraph beginning “most studies evaluating…” does not add significant value to the discussion in my opinion and could be omitted

Response: We agree with the Reviewer. This paragraph was removed from the Discussion section.

Comment 13. Page 17, paragraph starting “stakeholders participating in…” also does not add much value, although does relate to the paragraph at the top of the page on other physical barriers to participation noted by parents/adolescents and could be merged, or otherwise omitted

Response: We thank the Reviewer for his/her comment. The paragraph was removed from the Discussion.

Comment 14. The discussion overall summarizes many of the key findings well and engages with the broader literature on this topic. However, it could be strengthened by further reflection on the implications of these findings for future research, development of interventions, and/or the delivery of services and lifestyle programs for this population

Response: We thank the Reviewer for his/her suggestion. Accordingly, text was added at the end of the Discussion section:

Line 365, previous version:

As mentioned in our study, adolescents with cancer need to access activities and services adapted to their age and need to obtain information and clear answers to their questions, even after the treatment period has ended. Accordingly, in young adult survivors of childhood cancer, the main barriers to health care follow-up were the lack of knowledge related to late effects and of primary care physicians on survivorship health issues [85,86]. Given that it is well documented that survivors of AYA cancer are at higher risk for cardiometabolic complications [26], there is a pressing need for health promotion interventions tailored for this age group to optimize their motivation, engagement and participation. Continuing medical education as well as community teachings should be put in place to ensure that healthcare professionals and adolescent survivors are aware of their health risks. 

Paragraph, new version:

As mentioned in our study, adolescents with cancer need to access activities and services adapted to their age and need to obtain information and clear answers to their questions, even after the treatment period has ended. Accordingly, in young adult survivors of childhood cancer, the main barriers to health care follow-up were the lack of knowledge related to late effects and of primary care physicians on survivorship health issues [85,86]. Given that it is well documented that survivors of AYA cancer are at higher risk for cardiometabolic complications [26], there is a pressing need for health promotion interventions tailored for this age group to optimize their motivation, engagement and participation. Continuing medical education as well as community teachings should be put in place to ensure that healthcare professionals and adolescent survivors are aware of their health risks. 

Based on the study results, we recommend that future health promotion interventions for adolescents with cancer consider assuring the stability of the stakeholders during the intervention in order to promote the establishment of a relationship of trust. Remote interventions should be made available in addition to face-to-face interventions. The option of continuing the intervention after treatment has ended should also be offered. Moreover, initiatives should be put in place to encourage interactions with adolescents experiencing similar situations and with healthy peers in order to break isolation and to maintain a sense of normalcy. Also, during the development process of a health promotion intervention, we recommend including adolescent patients of different ages, with different diagnoses and at different stages in the treatment trajectory.  Finally, because adolescents tend to live in the present moment, the short-term benefits of a healthy lifestyle should be emphasized rather than the long-term health benefits.

Comment 15. I think another strength worth noting is that around half of parents were fathers, who are often underrepresented

Response: We thank the Reviewer for pointing out this strength. We added the following text and references:

Line 377, previous version:

A strength of our study is having consulted with stakeholders working in oncology, adolescents treated for cancer and parents which allowed us to gather a comprehensive perspective. Combining 3 methods of data collection (interviews, focus groups, written questionnaires) contributes to the robustness of the results. By focusing only on adolescents rather than children of all ages or AYAs, our study filled a gap in knowledge in the field of adolescent oncology. Our study has some limitations. It is possible that conducting the study in the hospital where adolescent participants received their treatments and where stakeholders work resulted in a desirability bias. There is a possibility of selection bias since adolescents participating in the VIE study may be more receptive to healthy lifestyle interventions.

Discussion, new version:

A strength of our study is having consulted with stakeholders working in oncology, adolescents treated for cancer and parents which allowed us to gather a comprehensive perspective. Combining 3 methods of data collection (interviews, focus groups, written questionnaires) contributes to the robustness of the results. By focusing only on adolescents rather than children of all ages or AYAs, our study filled a gap in knowledge in the field of adolescent oncology. Another strength is that half the parents were fathers, a population who is often underrepresented in qualitative studies involving parents of children with cancer (ref Sawyer 2017 and Williams 2015). Our study has some limitations. It is possible that conducting the study in the hospital where adolescent participants received their treatments and where stakeholders work resulted in a desirability bias. There is a possibility of selection bias since adolescents participating in the VIE study may be more receptive to healthy lifestyle interventions.

References to support changes made at comment 15:

Sawyer, S. M., McNeil, R., McCarthy, M., Orme, L., Thompson, K., Drew, S., & Dunt, D. (2017). Unmet need for healthcare services in adolescents and young adults with cancer and their parent carers. Supportive care in cancer : official journal of the Multinational Association of Supportive Care in Cancer25(7), 2229–2239. https://doi.org/10.1007/s00520-017-3630-y

Williams, L. K., & McCarthy, M. C. (2015). Parent perceptions of managing child behavioural side-effects of cancer treatment: a qualitative study. Child: care, health and development41(4), 611–619. https://doi.org/10.1111/cch.12188

Reviewer 2 Report

Dear authors,

The paper entitled "Needs, barriers and facilitators of adolescents participating in a lifestyle promotion program in oncology: stakeholders, adolescents and parents' perspective" is an exciting read. This paper highlights the needs, enablers and barriers to healthy lifestyle promotion interventions for adolescents and offers answers on adapting lifestyle promotion interventions. The introduction is strong; the results are well presented and adequately discussed. My comments are minor and are mainly related to methods.

a. Line 99: Could you please add more information on the VIE study? It helps the readers understand VIE a little before going to the web link.

b. Section 2.3: Is there any difference between stakeholders and VIE study stakeholders? If so, how did you differentiate?

c. I am a bit unclear about stakeholders' recruitment and data collection. Section 2.3.1 discusses focus groups with stakeholders, and 2.3.2 talks about self-report questionnaires to VIE study stakeholders. Did you also conduct a focus group with VIE stakeholders?

Author Response

Comments and Suggestions for Authors

Reviewer #2

Dear authors,

The paper entitled "Needs, barriers and facilitators of adolescents participating in a lifestyle promotion program in oncology: stakeholders, adolescents and parents' perspective" is an exciting read. This paper highlights the needs, enablers and barriers to healthy lifestyle promotion interventions for adolescents and offers answers on adapting lifestyle promotion interventions. The introduction is strong; the results are well presented and adequately discussed. My comments are minor and are mainly related to methods.

Comment a. Line 99: Could you please add more information on the VIE study? It helps the readers understand VIE a little before going to the web link.

Response: We thank the Reviewer for his/her encouraging comments regarding our study. According to the Reviewer’s comment, we have added the following text in the Methods section:

Paragraph starting at line 99, previous version:

This research is part of the VIE (Valorization, Implication, Education) study (web link: VIE Study) conducted at the CHU Sainte-Justine (CHUSJ) in Montreal (Quebec, Canada). The proposed integrative and personalized program informs newly diagnosed patients, their families, and healthcare professionals about the benefits of adopting early healthy lifestyles, particularly a balanced diet, physical activity (PA), and stress management. The design of the interventions has been described in detail [54,57,58]. The current work consists in a qualitative study conducted with a subgroup of the VIE study adolescent participants, in addition to their parents, and stakeholders working in oncology at CHUSJ (health care and research personnel). Participants were recruited from January 2018 until December 2021. Data were collected using focus groups, questionnaires, and interviews. This study was approved by the Ethics Review Committee of the CHUSJ (#2021-3129 and #2021-3278).

Paragraph, new version:

This research is part of the VIE (Valorization, Implication, Education) study (web link: VIE Study), a non-randomized controlled feasibility study conducted at the CHU Sainte-Justine (CHUSJ) in Montreal (Quebec, Canada). The proposed integrative and personalized program informs newly diagnosed patients and their families about the benefits of adopting early healthy lifestyles, particularly a balanced diet, physical activity (PA), and stress management to improve quality of life and prevent the incidence of late effects. The design of the interventions has been described in detail [54,57,58]. Briefly, participants 21 years old or younger at diagnosis; treated with chemotherapy or radiotherapy and able to give informed consent (by parents or legal guardians) were invited to participate. Patients were excluded if they were not being treated with chemotherapy and/or radiotherapy. Eligible participants were enrolled between the fourth and twelve weeks after cancer diagnosis. At recruitment, participants underwent an initial evaluation in nutrition, PA and psychological health. Follow-up visits were planned every month by alternating nutrition/PA consults. In parallel, the psychosocial support program was deployed in vulnerable families. Comprehensive nutrition, PA and psychological health evaluations were performed one year after the beginning of the intervention and after the end of cancer treatment. The control group was recruited from patients who had completed their treatment using the same criteria as for the protocol intervention group. No intervention was offered to the control group participants other than standard clinical follow-ups. The current work consists in a qualitative study conducted with a subgroup of the VIE study adolescent participants, in addition to their parents, and stakeholders working in oncology at CHUSJ (health care and research personnel). Participants were recruited from January 2018 until December 2021. Data were collected using focus groups, questionnaires, and interviews. This study was approved by the Ethics Review Committee of the CHUSJ (#2021-3129 and #2021-3278).

Comment b. Section 2.3: Is there any difference between stakeholders and VIE study stakeholders? If so, how did you differentiate?

Response: We agree with the Reviewer that this needs clarification. There were 2 different categories of stakeholders who participated in our study: 1) healthcare professionals and hospital-based teachers working with adolescents in oncology at CHUSJ who participated in the focus groups and 2) research health personnel who delivered the VIE study intervention, comprising of research kinesiologists and nutritionists who were eligible to answer the self-report questionnaire. Therefore, the stakeholders who participated in the focus group were not the same as those who completed the self-report questionnaire.

All survey questions were open-ended. Specifically, the questions were:

  • What are the specific needs of adolescents in an oncology setting?
  • What can negatively affect the participation of adolescents in lifestyle interventions?
  • What could be done to optimize the participation of adolescents in lifestyle interventions?
  • When do you think is the best time to initiate a lifestyle intervention for adolescents in oncology?

In the present article, we did not develop on the fourth question as it deviated from the aims.

According to the Reviewer’s comments, these changes have been performed in the Methods section – 2.2.1 Stakeholders:

Paragraph starting at line 113, previous version:

Stakeholders were eligible for participation in focus groups if they were: 1) a healthcare professional or a hospital-based teacher working with children and adolescents in oncology at CHUSJ and 2) exposed to the VIE study (health care personnel). Stakeholders were eligible to answer the written questionnaires if they: 1) were a member of the VIE research team, 2) had been working directly with participants in the nutrition or PA components study (research personnel). 

Paragraph, new version:

Two categories of stakeholders were included in the study: 1) healthcare professionals and hospital-based teachers working with adolescents in oncology at CHUSJ who participated in the focus groups and 2) research health personnel who delivered the VIE study intervention, comprising of research kinesiologists and nutritionists who were eligible to answer the self-report questionnaire. Stakeholders were eligible for participation in focus groups if they were: 1) a healthcare professional or a hospital-based teacher working with children and adolescents in oncology at CHUSJ and 2) exposed to the VIE study (health care personnel). Stakeholders were eligible to answer the self-report questionnaires if they: 1) were a member of the VIE research team, 2) had delivered the nutrition (nutritionists) or PA (kinesiologists) interventions. 

Comment c. I am a bit unclear about stakeholders' recruitment and data collection. Section 2.3.1 discusses focus groups with stakeholders, and 2.3.2 talks about self-report questionnaires to VIE study stakeholders. Did you also conduct a focus group with VIE stakeholders?

Response: We thank the Reviewer for his/her question. We hope that the response to his/her previous comment and the changes provided clarify this issue.
